# Ancestral origins are associated with SARS-CoV-2 susceptibility and protection in a Florida patient population

Yiran Shen[1], Bhuwan Khatri[2], Santosh Rananaware[3], Danmeng Li[4], David A. Ostrov[4], Piyush K. Jain[3], Christopher J. Lessard[2], Cuong Q. Nguyen[1,5,6]*

**1** Department of Infectious Diseases and Immunology, College of Veterinary Medicine, University of Florida, Gainesville, Florida, United States of America, **2** Genes and Human Disease Research Program, Oklahoma Medical Research Foundation, Oklahoma City, Oklahoma, United States of America, **3** Department of Chemical Engineering, University of Florida, Gainesville, Florida, United States of America, **4** Department of Pathology, Immunology & Laboratory Medicine, University of Florida, Gainesville, Florida, United States of America, **5** Department of Oral Biology, College of Dentistry, University of Florida, Gainesville, Florida, United States of America, **6** Center of Orphaned Autoimmune Diseases, University of Florida, Gainesville, Florida, United States of America

* nguyenc@ufl.edu

**Data Availability Statement:** The genotyping data been deposited in NCBI's Gene Expression Omnibus (GEO) and are accessible through GEO Series accession number GSE211972 (https://

## Abstract

COVID-19 is caused by severe acute respiratory syndrome-coronavirus-2 (SARS-CoV-2). The severity of COVID-19 is highly variable and related to known (e.g., age, obesity, immune deficiency) and unknown risk factors. The widespread clinical symptoms encompass a large group of asymptomatic COVID-19 patients, raising a crucial question regarding genetic susceptibility, e.g., whether individual differences in immunity play a role in patient symptomatology and how much human leukocyte antigen (HLA) contributes to this. To reveal genetic determinants of susceptibility to COVID-19 severity in the population and further explore potential immune-related factors, we performed a genome-wide association study on 284 confirmed COVID-19 patients (cases) and 95 healthy individuals (controls). We compared cases and controls of European (EUR) ancestry and African American (AFR) ancestry separately. We identified two loci on chromosomes 5q32 and 11p12, which reach the significance threshold of suggestive association ($p < 1 \times 10^{-5}$ threshold adjusted for multiple trait testing) and are associated with the COVID-19 susceptibility in the European ancestry (index rs17448496: odds ratio[OR] = 0.173; 95% confidence interval[CI], 0.08–0.36 for G allele; $p = 5.15 \times 10^{-5}$ and index rs768632395: OR = 0.166; 95% CI, 0.07–0.35 for A allele; $p = 4.25 \times 10^{-6}$, respectively), which were associated with two genes, PPP2R2B at 5q32, and LRRC4C at 11p12, respectively. To explore the linkage between HLA and COVID-19 severity, we applied fine-mapping analysis to dissect the HLA association with mild and severe cases. Using *In-silico* binding predictions to map the binding of risk/protective HLA to the viral structural proteins, we found the differential presentation of viral peptides in both ancestries. Lastly, extrapolation of the identified HLA from the cohort to the worldwide population revealed notable correlations. The study uncovers possible differences in susceptibility to COVID-19 in different ancestral origins in the genetic background, which may provide new insights into the pathogenesis and clinical treatment of the disease.

www.ncbi.nlm.nih.gov/geo/query/acc.cgi?acc=
GSE211972).

**Funding:** The study is supported financially in part
by PHS grants DE028544 and DE028544-02S1
from the National Institute of Dental and
Craniofacial Research. The funders had no role in
study design, data collection and analysis, decision
to publish, or preparation of the manuscript.

**Competing interests:** The authors have declared
that no competing interests exist.

## Introduction

In December 2019, a novel coronavirus, severe acute respiratory syndrome-coronavirus-2 (SARS-CoV-2), emerged in Wuhan, Hubei Province, China, initiating a breakout of atypical acute respiratory disease, termed coronavirus disease 2019 (COVID-19). SARS-CoV-2 is a *betacoronavirus* in the family of *Coronaviridae*; the virus contains four structural proteins: S (spike), E (envelope), M (membrane), and N (nucleocapsid), sixteen non-structural proteins (nsp1–16) and eleven accessory proteins, which support viral essential physiological function and evasion from the host immune system [1]. The complex structure of the virus provides multiple possible targets for antiviral prevention and treatment; however, the lack of comprehensive knowledge of viral infection and host immune response has hampered efforts to predict the disease course and identify effective therapeutic candidates. One of the most striking features of SARS-CoV-2 is consequence variability, ranging from asymptomatic to symptomatic viral pneumonia and finally to life-threatening acute respiratory distress syndrome [2]. A majority of patients recovered during early infection, but a smaller percentage of patients were more likely to progress and eventually die from severe systemic inflammatory response syndrome. Several factors are associated with disease severity, e.g., age, gender, pre-existing conditions [3, 4] and race [5–7].

To reveal the underlying pathogenesis of SARS-CoV-2 susceptibility and disease progression, genome-wide association studies (GWAS) provide additional clues regarding the pathogenesis of complex diseases by identifying potential susceptible allelic variants. Several loci on the different chromosomes have been previously reported to be associated with COVID-19 severity [8–10]. Some studies have focused on the genetic linkage between HLA alleles and SARS-CoV-2 infection. The class I (HLA-A, -B, and -C) and class II HLA (HLA-DR, -DQ, and -DP) exhibit a high degree of polymorphism, and CD4$^+$ T cells and CD8$^+$ T cells respond to pathogens by recognizing different classes of HLA molecules (I or II, respectively) on the cell surface. Specific HLA genotypes have been associated with T-cell mediated immunity and viral clearance. Several class I and II alleles have been identified to be related to SARS-CoV-2 infection, protection, and severity through *in silico* prediction [11–14], patient genotyping [15–18], and whole-genome sequencing [19]. These studies suggested that genetic variants, especially HLA alleles, were associated with disease morbidity, mortality, and prognosis.

To study the COVID-19 consequences variability, we applied GWAS on 284 SARS-CoV-2 positive samples and 89 negative samples composed of different ancestry origins to determine if a specific genetic factor was associated with susceptibility to SARS-CoV-2 infection and severity of COVID-19 on different genetic background. We also applied fine-mapping to reveal potential disease-associated HLA alleles in European and African ancestral populations. In addition, we applied *in silico* prediction and structural modeling to identify and map the structural epitopes presented by the associated protective and risk HLAs. Lastly, we extrapolated the finding to the worldwide population and the result showed significant correlations with other countries.

## Materials and methods

### Study population

284 confirmed COVID-19 samples were obtained from Boca Biolistics (Pompano Beach, FL) and CTSI Biorepository at the University of Florida (Gainesville, FL). The median age of the patients was 44.7 years (range: 3–94 years). Patients had positive test results for SARS-CoV-2 by RT-PCR from nasopharyngeal swabs, saliva, or tracheal aspirates. 89 healthy individuals with negative PCR tests for SARS-CoV-2 viral infection were included as controls. The median

age of the control group was 60 years (range: 0-101 years) (**S1 Table**). After the exclusion of samples during quality control, the final case-control data sets comprised 254 patients and 80 control participants. The study was approved by the Institutional Review Board of the University of Florida. The written informed consent was collected from each participant prior to the start of the sampling.

## Sample extraction and genotyping

Clinical specimens of nasopharyngeal swabs, saliva, and tracheal aspirates were collected in a viral transport medium. DNA was extracted from viral transport medium or directly from tracheal aspirates by Maxwell® RSC Blood DNA Kit per manufacturer's instructions (Promega Corporation). RNase A was added to samples to remove potential viral RNA. Isolated genomic DNA was quantified by NanoDrop™ One/OneC Microvolume UV-Vis Spectrophotometer (Thermo Scientific). Genotyping was done using Axiom™ Human Genotyping SARS-CoV-2 Research Array as instructed by the manufacturer (Thermo Scientific).

## Quality control

PLINK (v1.9) [20] was used for quality control and logistic analysis of the data. The single nucleotide polymorphism (SNPs) and subjects passing the following quality control criteria [21] were used in the downstream analysis: SNPs having Major Allele Frequency >1%, SNPs and sample each with call rate >95%, controls with Hardy-Weinberg proportion test with p>0.001 and cases and controls with differential missingness P>0.001, subjects with heterozygosity (<5 S.D. from the Mean), and one individual from the pair was removed if identity-by-descent (IBD) was >0.4.

## Assessment of population stratification

Principal components between cases and controls and population substructures within the dataset were determined using EIGENSTRAT [22] and independent genotyped SNPs with $r^2 < 0.2$ between variants, for this 1000 Genome reference population was used. Principle component analysis (PCA) was used to remove outliers defined by standard deviations greater than 6 (s.d. >6) from the Mean [21]. Case and control samples were plotted by PC1 and PC2.

## Imputation

Whole-genome imputation was performed using TOPMed Reference Panel (TOPMed r2) in the TOPMed Imputation server [23]. The data were phased using Eagle version 2.4 and imputed using Minimac4. In addition, the HLA (chr6) region was imputed in Michigan Imputation Server [23]. The data were phased using Eagle V2.4 and imputed using Minimac4 and Four-digit Multi-ethnic HLA reference panel.

## Logistic analysis

Post imputation, a quality control measure as explained before, was used in the imputed data. Logistic regression analysis was carried out using PLINK to test for single marker SNP-COVID-19 association post imputation, adjusting for the first two principal components.

### *In-silico* binding predictions

To predict the SARS-CoV-2 peptides in which selective HLA alleles will bind, HLA peptide-binding prediction algorithms netMHCpan (v4.1) and netMHCIIpan (v4.0) were utilized for HLA class I and class II alleles, respectively [24]. Full-length amino acid sequences of structural

proteins from SARS-CoV-2 whole-genome proteome (SnapGene, EPI_ISL_7196120_B.1.1.529, EPI_ISL_7196121_B.1.1.529) were used to infer all possible potentially relevant peptides (9mers for class I and 15mers for class II). Default rank thresholds (%Rank values) were used to define strong (0.5% for netMHCpan and 2% for netMHCIIpan) and weak (2% for netMHCpan and 10% for netMHCIIpan) binders. The prediction binding pattern was compared among different alleles to predict the immunogenic portion for further application. In selecting the peptides of interest, %Rank was used as a reference value to compare the ability of the same peptide to be presented by risk and protective alleles among predicted alleles that can be presented by at least one of the candidate alleles (Class I MHC: %Rank < 2%, Class II MHC: %Rank < 10), the top three peptides have the greatest difference in presentation ability (%Rank) from the other allele are selected.

## Structural modeling of SARS-CoV-2 peptide HLA molecules interactions

Models for all the HLA molecules were made using SWISS-MODEL. 7RTD was used as templates for HLA-B*27:05 and HLA-C*13:02, and 3PDO for HLA-DRB1*13:02. 6PX6 and 5KSA were used as templates for HLA-DQA1*05:01 and HLA-DQB1*03:01, respectively. Then, they were superposed into 5KSU and merged into one final model for the HLA-DQ structure in COOT. The peptide in the template structure was mutated in COOT, too. The geometry of the result complexes was regularized in PHENIX.

## Worldwide geographical comparison

The data of allele frequencies among countries/regions were obtained from the Allele Frequency Net Database (http://allelefrequencies.net), global susceptibility among countries/regions (cases per one million population) were obtained from Worldometer (https://www.worldometers.info), and global death rate (case-mortality) were obtained from John Hopkins Coronavirus resource center (https://coronavirus.jhu.edu/data/mortality). Databases were searched on March 24th, 2022. When multiple data points of an allele frequency were available for a country, the weighted Mean was calculated according to the sample size.

## Statistical analysis

The association between the allele frequency of each HLA gene and cases and mortality were assessed by linear regression in GraphPad Prism (v9.3.1). An adjusted p-value of <0.05 was considered statistically significant.

## Results

### GWAS analysis of COVID-19 patients with European and African ancestries

The state of Florida in the United States (US) is one of the most ethnically diverse states. 53% of Floridians are White (Non-Hispanic), with 21.6% being White (Hispanic), Black or African American (Non-Hispanic) (15.2%), Asian (Non-Hispanic) (2.73%), and Other (Hispanic) (2.97%). 20.1% of Floridians were born outside the US, which is higher than the national average of 13.7% in 2019 (https://datausa.io/profile/geo/florida). Due to the ethnically diverse nature of Florida residents, we performed QC of the dataset as described previously in methods [21]. Following QC, we stratified the population based on their genetic ancestry (COVID-19 cases: 56% European (EUR) ancestry and 37% African (AFR) ancestry. Non-COVID-19 controls: 77% EUR ancestry and 15% AFR ancestry (**S1 and S2 Figs**). Since EUR and AFR ancestries are distinct populations, we performed GWAS analysis on the two populations

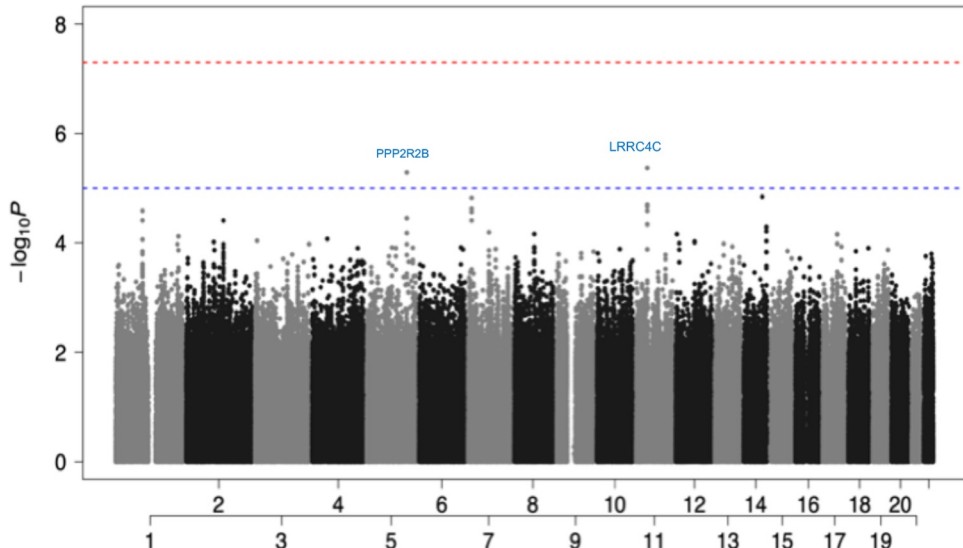

**Fig 1. Manhattan plot showed two protectivity loci with suggestive significance in EUR ancestry.** Two loci were found to be associated with COVID-19 susceptibility reach the significance threshold of suggestive association (P < 1 x $10^{-5}$ threshold adjusted for multiple trait testing) in the EUR group: the rs17448496 at locus 5q32 showed a suggestive association with serine/threonine-protein phosphatase 2A regulatory subunit B (PPP2R2B), and the rs768632395 at locus 11p12 showed a suggestive association with Leucine Rich Repeat Containing 4C (LRRC4C). (Red: GWAS association (P < 5 x $10^{-8}$); Blue: suggestive association (P < 1 x $10^{-5}$)).

separately. As presented in **Fig 1**, we found top variants rs17448496 at locus 5q32 (odds ratio [OR] = 0.173; 95% confidence interval[CI], 0.08–0.36 for G allele; p<5.15x$10^{-6}$) and rs768632395 at locus 11p12 (OR = 0.166; 95% CI, 0.07–0.35 for A allele; p<4.25 ×$10^{-6}$) that were suggestive GWAS with COVID-19 susceptibility in the EUR ancestral patients. There were no SNP association signals in the AFR ancestral patients that met the significance threshold of suggestive association (**Fig 2**). In summary, the limited sample size was sufficient to distinguish two different genetic ancestries. In addition, we identified two interesting SNPs

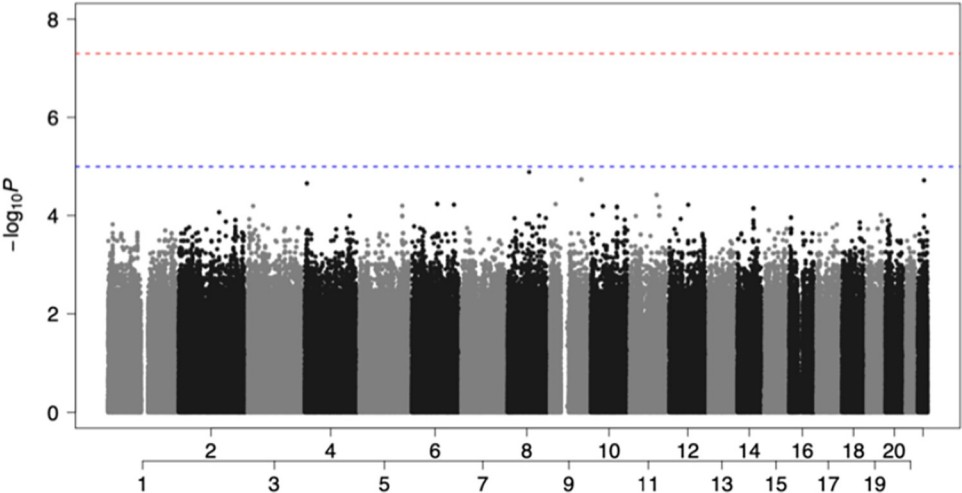

**Fig 2. Manhattan plot showed non-significant associated loci in AFR ancestry.** There were no SNP association signals in the AFR group that met the significance threshold of suggestive association: P<1×$10^{-5}$. (Red: GWAS association (P < 5 x $10^{-8}$); Blue: suggestive association (P < 1 x $10^{-5}$)).

localized in the EUR ancestry. These two SNPs showed OR values of less than 1, suggesting that they are possibly involved to some extent in protection against SARS-CoV-2 infection (i.e., to a greater extent, would be present in unconfirmed healthy individuals) in contrast to the AFR ancestral patients where the absence of prominent SNPs, which can be interpreted as a protective mechanism with no apparent contribution to SARS-CoV-2 infection.

**Chromosome 5q32 and 11p12.** As presented in **Fig 1**, the GWAS showed a suggestive association with serine/threonine-protein phosphatase 2A (PP2A) regulatory subunit B (PPP2R2B) (rs17448496, OR = 0.173; 95% CI = 0.08–0.36 for G allele; p<$5.15 \times 10^{-6}$)) on chromosome 5q32, and Leucine-Rich Repeat Containing 4C (LRRC4C) (rs768632395, OR = 0.166; 95% CI = 0.07–0.35 for A allele; p<$4.25 \times 10^{-6}$) on chromosome 11p12. PPP2R2B belongs to the phosphatase family, and they are involved in several biological processes. Previous reports have shown that PP2A can activate T-cell responses by inhibiting cytotoxic T-lymphocyte-associated (CTLA)-4 function or impairing programmed death-ligand (PD-L)-1 expression [25]. The PPP2R2B gene can encode the regulatory subunit B55β, forming the PP2A-B55β complex by binding to the scaffolding and catalytic subunits. PPP2R2B plays an important regulatory role in the immune system, preventing organ damage by activated T cells in chronic inflammation caused by systemic autoimmune diseases, hypermethylation of PPP2R2B can induce defective acquired apoptosis [26], and dysregulation of PPP2R2B may contribute to the development and progression of breast cancer [27]. Importantly, PPP2R2B interacted with PPP1R15A in the ERK signaling pathway, which is on chr9 and was associated with SARS--CoV-2 infection in the C5 phenotype from HGI (COVID-19 Host Genetics Initiative group) [28]. LRRC4, also known as netrin-G ligand-2 (NGL-2), belongs to the superfamily of LRR proteins and is a receptor for netrin-G2 [29], regulates excitatory synapse formation and promotes axonal differentiation. LRRC4 can also act as a tumor suppressor gene to significantly inhibit glioblastoma cell proliferation by interacting with extracellular and intracellular signaling pathways [30]. LRRC4 binds to phosphotyrosine-dependent protein kinase 1 (PDPK1), promotes NF-κB activation in glioblastoma cells and secretion of Interleukin 6 (IL-6), C-C Motif Chemokine Ligand 2 (CCL2) and Interferon-gamma (IFN-γ), thereby inhibiting the expansion of tumor-infiltrating regulatory T cells and the growth of glioblastoma cells [31]. Although not as significant as the findings of the previous meta-analysis, the loci we observed in the EUR ancestral patients were associated with host-adaptive, especially T-cell-related immunity, and the lack of such a significance in AFR ancestral patients may explain the susceptibility of the population.

## HLA association in COVID-19 patients with European and African ancestries

Specific HLA alleles have played significant roles in many bacterial and viral infections. We applied the fine-mapping to the extended HLA region (chromosome 6, 25 to 34 Mb) to determine their association with the EUR and AFR ancestral patients. We determined no significant SNP association signal on the HLA complex that achieved the threshold of significance for suggestive association (**Figs 1 and 2**). Due to the substantial overlap in bound peptides among HLA alleles and their co-dominant expression, additive GWAS association tests may not capture the full functional role of HLA in COVID-19 risk. Therefore, we further analyzed the association of COVID-19 with HLA alleles through HLA imputation. The results showed multiple alleles present for HLA-A, -B, -C, -DPA1, -DPB1, -DQA1, -DQB1, -DRB1 loci through imputation-based methods using the SNPs data from GWAS. Indeed, it is well established that minority groups of different races and ethnic groups in the US are disproportionately affected by COVID-19. Minorities endured a higher risk for infection, hospitalization, and death [6,

**Table 1. Significant alleles associate with either protective (OR <1) or risk (OR>1) factor overall in EUR and AFR group.**

| Allele | Odds Ratio | Standard Error | L95[a] | U95[b] | P value |
|---|---|---|---|---|---|
| **EUR** | | | | | |
| HLA-B*27:05 | 0.17 | 0.81 | 0.03 | 0.84 | 0.0292 |
| **AFR** | | | | | |
| HLA-A*02:01 | 0.20 | 0.69 | 0.05 | 0.80 | 0.0228 |
| HLA-A*33:01 | 0.05 | 1.28 | 0.005 | 0.71 | 0.0259 |
| HLA-DRB1*13:02 | 0.19 | 0.67 | 0.05 | 0.72 | 0.0141 |
| HLA-DPB1*11:01 | 0.16 | 0.77 | 0.03 | 0.72 | 0.0167 |

[a] Lower 95 confidence interval

[b] Upper 95 confidence interval

32]. The allele distributions were compared between COVID-19 patients and control individuals in EUR and AFR ancestries, respectively. As presented in **Table 1**, significant associations (p≤0.05) between HLA-B*27:05 alleles and SARS-CoV-2 positivity were identified in the EUR ancestry, which was associated with a decreased risk of SARS-CoV-2 positivity (OR = 0.17; 95% CI, 0.03–0.83; p = 0.029). In the AFR ancestry, HLA-A*02:01 (OR = 0.20; 95% CI, 0.05–0.80; p = 0.023), -A*33:01 (OR = 0.05; 95% CI, 0.005–0.71; p = 0.026), -DRB1*13:02 (OR = 0.19; 95% CI, 0.05–0.72; p = 0.014) and -DPB1*11:01 (OR = 0.16; 95% CI, 0.03–0.72; p = 0.023) were associated with a decreased risk of SARS-CoV-2 positivity. Overall, individuals who carry the above class I and class II HLA alleles were less likely to be infected by SARS-CoV-2.

## HLA association between severe and mild COVID-19 patients

It is well documented that COVID-19 patients exhibited a whole range of symptoms and severity. To further define the HLA association with the severity of COVID-19, we subdivided the patient cases into mild and severe according to the source of sample acquisition (saliva, nasopharyngeal swabs: mild; tracheal aspirates: hospitalized/severe) and performed the same HLA imputation. Stratified by disease severity showed significant alleles changed in both EUR and AFR ancestral patient populations. In the EUR ancestral patients, HLA-C*12:03, -B*35, -B*38:01 were associated with an increased risk of SARS-CoV-2 severity (OR >1) (**Table 2**). Interestingly, HLA-B*38 showed a significant odds ratio (OR = 1639). Whereas, as presented in **Table 3**, in the AFR ancestral patients, HLA-DQB1*03 was associated with an increased risk of SARS-CoV-2 severity. In contrast, HLA-B*58, -DRB1*13:02, and -DQB1*06 were associated with decreased risk of SARS-CoV-2 severity. The results suggest that there is a higher frequency of risk and protective alleles in mild and severe cases of different ancestral patient

**Table 2. Significant alleles associate with either protective (OR <1) or risk (OR>1) factor in EUR group by case severity.**

| Allele | Odds Ratio | Standard Error | L95 | U95 | P value |
|---|---|---|---|---|---|
| **Severe vs Control** | | | | | |
| HLA-C*12:03 | 8.07 | 0.94 | 1.28 | 50.85 | 0.0261 |
| HLA-B*35 | 4.21 | 0.71 | 1.04 | 17.04 | 0.0441 |
| **Severe vs Mild** | | | | | |
| HLA-B*38:01 | 1639.00 | 3.65 | 1.29 | 2076000 | 0.0423 |
| **Mild vs Control** | | | | | |
| HLA-A*11:01 | 0.24 | 0.71 | 0.06 | 0.95 | 0.0426 |

**Table 3. Significant alleles associate with either protective (OR <1) or risk (OR>1) factor in AFR group by case severity.**

| Allele | Odds Ratio | Standard Error | L95 | U95 | P value |
|---|---|---|---|---|---|
| **Severe vs Control** | | | | | |
| HLA-DQB1*03 | 115.80 | 1.96 | 2.50 | 5360.00 | 0.0151 |
| HLA-B*58 | 0.07 | 1.10 | 0.01 | 0.61 | 0.0159 |
| HLA-DRB1*13:02 | 0.09 | 1.14 | 0.01 | 0.89 | 0.0393 |
| **Severe vs Mild** | | | | | |
| HLA-DQB1*06 | 0.41 | 0.39 | 0.19 | 0.88 | 0.0228 |
| **Mild vs Control** | | | | | |
| HLA-DQA1*02:01 | 0.13 | 0.89 | 0.02 | 0.76 | 0.0233 |
| HLA-DRB1*07:01 | 0.15 | 0.85 | 0.03 | 0.80 | 0.0258 |
| HLA-DPB1*11:01 | 0.06 | 1.29 | 0.01 | 0.81 | 0.0335 |
| HLA-A*02:01 | 0.23 | 0.74 | 0.05 | 0.97 | 0.0452 |
| HLA-DRB1*13:02 | 0.26 | 0.68 | 0.07 | 0.99 | 0.0489 |

populations, and this frequency could explain the demographic differences in the affected population.

### *In silico* mapping of peptide epitopes derived from SARS-CoV-2 structural proteins presented by protective and risk alleles in EUR ancestry

As presented in **Tables 1 and 2**, the protective and risk alleles of the EUR ancestry encode class I MHC molecules. To map which viral epitopes are presented by the protective and risk HLAs, we applied the prediction tool NetMHCpan—4.1 provided by DTU Health. As a model, we chose HLA-B*27:05 (protective) and HLA-C*12:03 (risk) based on the significance and mapped the SARS-CoV-2 structural proteins (S, M, N, and E). Both stronger (%Rank less than 0.5%) and weaker (%Rank less than 2%) binders of HLA-B*27:05 (46 peptides in total) and HLA-C*12:03 (151 peptides in total) were selected and mapped across the structural proteins (**Fig 3**). Based on the presented differences in protective and risk alleles, the top three peptides that have the most significant difference in presentation ability (%Rank) from the other allele are selected and highlighted in **Fig 3** and the sequences are listed in **Table 4**. Molecular docking combined with *in silico* mapping showed for peptides derived from S protein, protective allele HLA-B*27:05 presents ARDLICAQK and RRARSVASQ, corresponding to amino acid (aa) positions 846–854 and 682–690 in the spike protein S2 subunit, and KHTPINLVR (aa 206–214) from N-terminal domain (NTD) (**Fig 4A–4C**). Risk allele HLA-C*12:03 presents LAATKMSEC (aa 1024–1032) and IAPGQTGKI (aa 410–418) at S2 region, and FASTEKSNI (aa 92–100) from RBD (**Fig 4D–4F**). Notably, the protective allele failed to present antigen from E protein, and the risk allele shows peptides such as LTALRL-CAY (aa 32–42), LAFVVFLLV (aa 21–29) and LAILTALRL (aa 31–39) across the E protein (**Fig 4G–4I**). Similar analyses were performed to determine whether HLA-B*27:05 and HLA-C*12:03 can also present the same set of structural proteins for other variants. The results indicated that similar sequences of peptide antigens of delta and omicron variants were predicted to be presented by these alleles (**S2 Table**). Considering that HLA-B*38:01 has a large odds ratio (OR = 1639.00), we also did the same analysis (**S3 Table**) (**S3 and S4 Figs**) with HLA-B*38:01 as risk allele compared to HLA-B*27:05 (protective). There was no obvious difference in the pattern and type of antigen presentation preference between the two alleles, as the presentation of S proteins was concentrated in the S2 region, and both alleles were unable to present E proteins. In summary, the results demonstrated that in the EUR ancestry group,

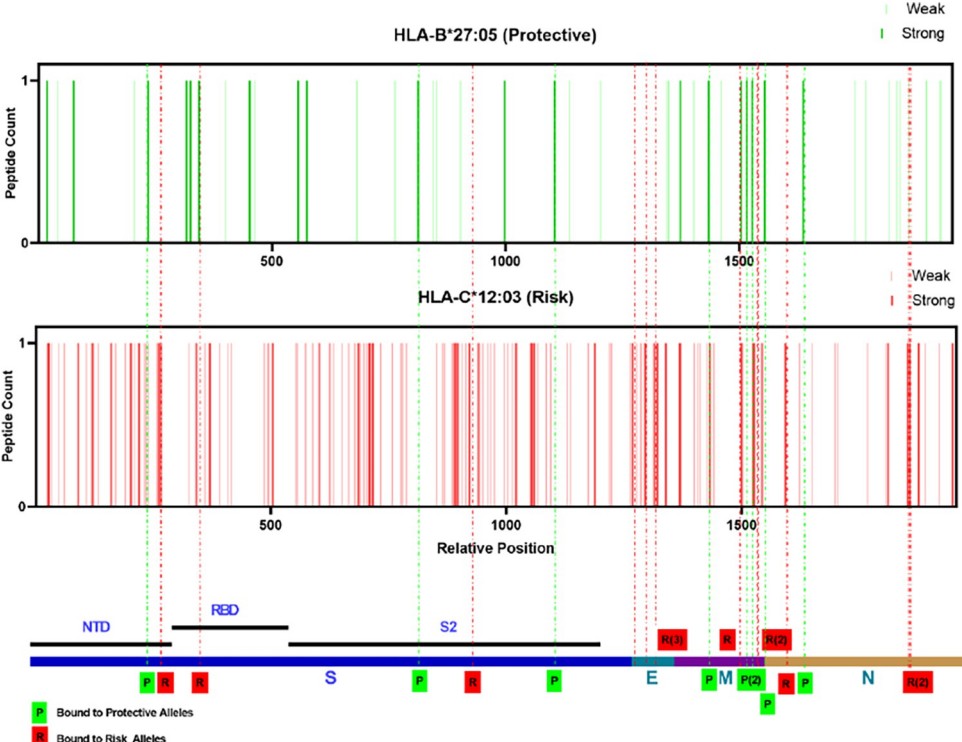

**Fig 3. Distribution of allelic presentation of peptide across the SARS-CoV-2 structural for protective (HLA-B\*27:05) and risk (HLA-C\*12:03) alleles in EUR ancestry group.** Dark and light bars indicating the identified stronger (< 0.5%Rank) and weaker (< 2%Rank) binding 9 mer peptides, respectively. With green and red indicating protective and risk alleles association, respectively. Dashed lines are the selected top three peptides in each structural protein that were presented with the greatest variation in binding affinity, marked by the final tendency of alleles (green: protective allele; red: risk allele). The relative positions are arranged by successive structural proteins in the order of S, E, M, N and relative lengths as indicated in the bottom.

HLA-B\*27:05 (protective) and HLA-C\*12:03 (risk) present multiple structural proteins of SASR-CoV-2 and the protective alleles lack presentation of the E protein.

## *In silico* mapping of antigen derived from SARS-CoV-2 structural protein presented by protective and risk alleles in AFR ancestry

The most protective and risk alleles were class II HLA in AFR ancestry, unlike the EUR ancestry (Table 3). To determine which viral epitopes are presented by the protective and risk HLAs, we again applied the prediction tool NetMHCIIpan—4.0. Prediction based on HLA-DQ molecules requires both α and β chains, we chose HLA- DQA1\*05:01 (not significant in HLA imputation) as the α-chain based on the overall haplotype frequency and predictive accuracy. As a model, we chose HLA-DRB1\*13:02 (protective) and HLA-DQA1\*05:01-DQB1\*03:01 (risk) and mapped them for SARS-CoV-2 structural proteins. Both stronger (% Rank less than 2%) and weaker (%Rank less than 10%) binders of HLA-DRB1\*13:02 (248 peptides in total) and HLA-DQB1\*03:01 (206) were selected and mapped across the structural proteins (Fig 5). Again, the top three peptides that have the most significant difference in presentation ability (%Rank) from the other allele were selected and highlighted in Fig 5, and the sequences were listed in Table 5. The molecular docking combined with *in silico* mapping showed for peptides derived from S protein, protective allele HLA-DRB1\*13:02 tends to bind PRTFLLKYNENGTIT (aa 272–286) at NTD, KKSTNLVKNKCVNFN (aa 528–542) and

**Table 4. In silico binding prediction for protective (HLA-B*27:05) or risk (HLA-C*12:03) alleles in EUR group present structural protein of SARS-CoV-2 original strain.**

| Peptide | Protein | Bound Preference |
| --- | --- | --- |
| ARDLICAQK | S | Protective Allele |
| RRARSVASQ | S | Protective Allele |
| KHTPINLVR | S | Protective Allele |
| LAATKMSEC | S | Risk Allele |
| IAPGQTGKI | S | Risk Allele |
| FASTEKSNI | S | Risk Allele |
| LTALRLCAY | E | Risk Allele |
| LAFVVFLLV | E | Risk Allele |
| LAILTALRL | E | Risk Allele |
| YRINWITGG | M | Protective Allele |
| KKLLEQWNL | M | Protective Allele |
| SRYRIGNYK | M | Protective Allele |
| IAIAMACLV | M | Risk Allele |
| AAVYRINWI | M | Risk Allele |
| LAAVYRINW | M | Risk Allele |
| RRIRGGDGK | N | Protective Allele |
| DRLNQLESK | N | Protective Allele |
| GRRGPEQTQ | N | Protective Allele |
| FAPSASAFF | N | Risk Allele |
| SAFFGMSRI | N | Risk Allele |
| LSPRWYFYY | N | Risk Allele |

KSTNLVKNKCVNFNF(aa 529–543) at RBD (**Fig 6A–6C**). While risk allele HLA-DQB1*03:01 prefers to bind ITPCSFGGVSVITPG (aa 587–601), ECDIPIGAGICASYQ (aa 661–675) and TPCSFGGVSVITPGT (aa 588–602) at beta-strand region on S2 (**Fig 6D and 6E**), which is less immunogenetic, thus differences in the disease course may be the result of a combination of multiple alleles. Interestingly, different from the EUR group, only the protective allele presents the peptide such as LVKPSFYVYSRVKNL (aa 1–65), VYSRVKNLNSSRVPD (aa 58–72), and SFYVYSRVKNLNSSR (aa 55–69) from E protein (**Fig 6F–6H**). In contrast, the risk allele failed to present peptide from E protein, leading to incomplete viral clearance and the subsequent induction of new mutations. We analyzed the antigen presentation of the same set of alleles for the delta and omicron variants (**S4 Table**) structural proteins. We found that this property was still retained. In summary, the result demonstrated that in the AFR ancestry group, the antigens presented by protective alleles (HLA-DRB1*13:02) are more diverse, with different presentation sites against the S, M, and N proteins. In contrast, the antigens presented by risk alleles (HLA-DQB1*03:01) are more at a single site and lack presentation of the E protein.

## Association of the identified HLA alleles with worldwide COVID-19 cases and mortalities

To further evaluate whether the identified risk and protective HLA in the studied cohort in the state of Florida can be extrapolated to determine the association with cases and mortalities worldwide, we employed linear regression to determine the association of each allele frequency, number of COVID-19 cases per one million population, and case-mortality due to COVID-19. In alleles predicted by disease severity (**Tables 2 and 3**), We selected two

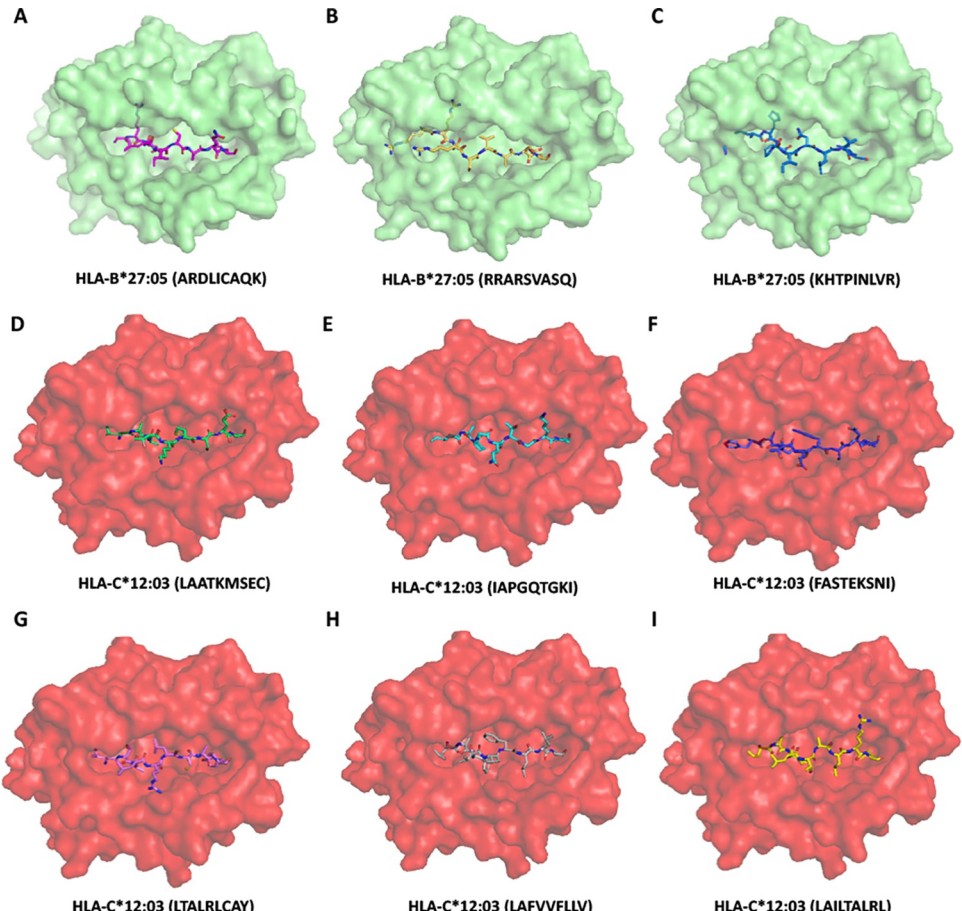

**Fig 4. Model of class I HLA molecule associated with protective (HLA-B*27:05) or risk (HLA-C*12:03) incidence of SARS-CoV-2 severity.** (A-C) HLA-B*27:05 binds to ARDLICAQK, RRARSVASQ and KHTPINLVR derived from the S protein with affinity estimated Kd 1534 nM, 438.61 nM, 6798.81 nM respectively. (D-F) HLA-C*12:03 binds to LAATKMSEC, IAPGQTGKI and FASTEKSNI derived from the S protein with affinity estimated Kd 959.47 nM, 806.95 nM, 126.76 nM respectively. (G-I) HLA-C*12:03 binds to LTALRLCAY, LAFVVFLLV and LAILTALRL derived from the E protein with affinity estimated Kd 176.42 nM, 190.66 nM, 112.95 nM respectively. Peptides are shown as sticks modeled on the crystal structure of class I HLA molecues, different peptide backbones are represented by different colors.

protective alleles (A*11:01 and DPB1*11:01) and two risk alleles (B*38:01 and DQB1*03) based on their ORs. As presented in **Fig 7**, the protective allele A*11:01 showed a significant association. As the allele frequencies increase, the lesser case frequency per one million population (p = 0.0267); however, there was no significant correlation with case mortality. There was no statistical significance with the protective DPB1*11:01 allele for case mortality and case/1M pop (**Fig 7A**). Interestingly, when we evaluated the risk B*38:01 allele, which has the largest odds ratio (OR = 1639.00), it showed a positive or upward trend between the increase in the allele frequencies and rise in both case/1M population (p = 0.0101). Although the trends were similar, this was not significant in the mortality correlation (p = 0.6092) (**Fig 7B**). The risk allele DQB1*03 showed a significant association with case-mortality (p = 0.0099) but no significant correlation with case/1M population (p = 0.0782). The results suggest that extrapolation of specific protective and risk HLA alleles identified in the studied AFR and EUR cohort can be applicable to determine the association with worldwide COVID-19 cases and mortalities.

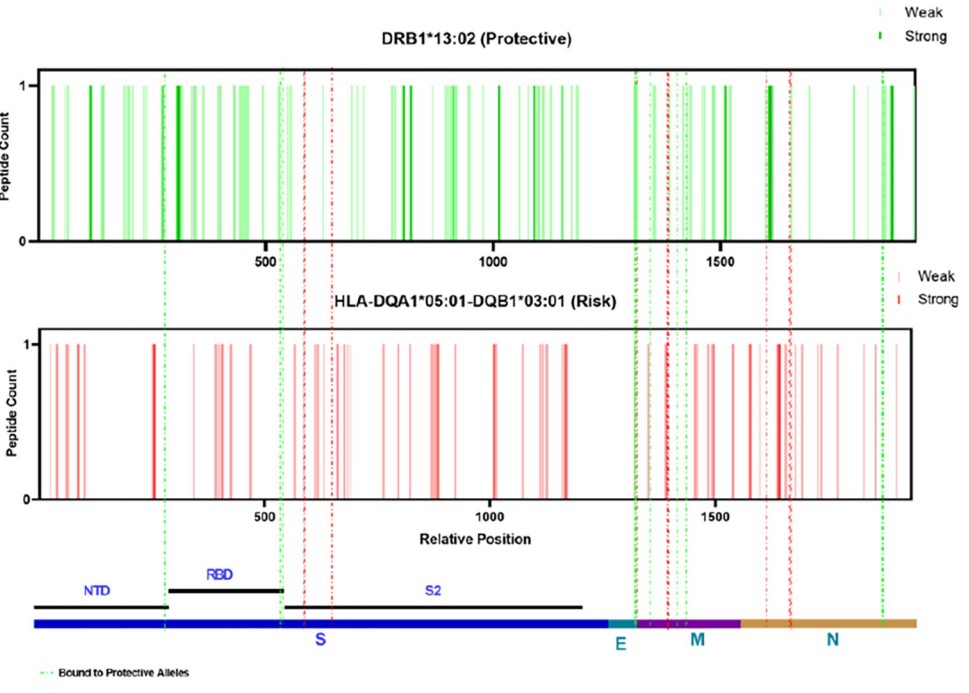

**Fig 5. Distribution of allelic presentation of peptide across the SARS-CoV-2 structural for protective (HLA-DRB1\*13:02) and risk (HLA-DQB1\*03:01) alleles in AFR ancestry group.** Dark and light bars indicating the identified stronger (< 2%Rank) and weaker (< 10%Rank) binding 12 mer peptides, respectively. With green and red indicating protective and risk alleles association, respectively. Dashed lines are the selected top three peptides in each structural protein that were presented with the greatest variation in binding affinity, marked by the final tendency of alleles (green: protective allele; red: risk allele). The relative positions are arranged by successive structural proteins in the order of S, E, M, N and relative lengths as indicated in the bottom.

## Discussion

In this study, the GWAS study showed two plausible genome-wide significant associations on chromosomes 5q32 and 11p12 in EUR ancestral group in our overall susceptibility model by ancestral stratification. Using HLA imputation, the results suggest that several class I MHC alleles in EUR ancestral group and a mixed class I and II MHC alleles in AFR ancestral group were likely to be associated with disease susceptibility or severity. For example, in EUR ancestral group, HLA-B\*27:05 were associated with an overall decreased risk of infection, HLA-A\*11:01 were associated with less severity, while HLA-C\*12:03, -B\*35, -B\*38:01 were associated with an increased risk of SARS-CoV-2 severity. In AFR ancestral group, HLA-A\*02:01, -DRB1\*13:02, and -DPB1\*11:01 were associated with an overall decreased risk of SARS-CoV-2 positivity. Regarding disease severity, HLA-DRB1\*13:02 and -DPB1\*11:01, together with HLA-B\*58, -DQB1\*06, -DQA1\*02:01, -DRB1\*07:01 exhibited an association with less severity, whereas HLA-DQB1\*03 was associated with an increased risk of SARS-CoV-2 severity. Using *in silico* prediction and modeling of SARS-CoV-2 structural proteins to identify the epitopes and binding strength with risk and protective HLA alleles showed that a different presentation pattern may activate the immune response to varying degrees, leading to changes in the course of the disease. Overall, the study sheds important insight into COVID-19 by stratifying ancestral origin, leading to better disease understanding and prevention strategies.

The first reported genome-wide association signals were at loci 3p21.31 and 9q34.2, which revealed the association of protein-coding genes that regulate viral attachment and host

**Table 5. In silico binding prediction for protective (HLA-DRB1\*13:02) or risk (HLA-DQA1\*05:01-DQB1\*03:01) alleles in AFR group present structural protein of SARS-CoV-2 original strain.**

| Peptide | Protein | Core bound protective | Core bound risk | Bound Preference |
|---------|---------|-----------------------|-----------------|------------------|
| PRTFLLKYNENGTIT | S | LKYNENGTI | LKYNENGTI | Protective Allele |
| KKSTNLVKNKCVNFN | S | LVKNKCVNF | LVKNKCVNF | Protective Allele |
| KSTNLVKNKCVNFNF | S | LVKNKCVNF | LVKNKCVNF | Protective Allele |
| ITPCSFGGVSVITPG | S | FGGVSVITP | SFGGVSVIT | Risk Allele |
| ECDIPIGAGICASYQ | S | IGAGICASY | IGAGICASY | Risk Allele |
| TPCSFGGVSVITPGT | S | FGGVSVITP | SFGGVSVIT | Risk Allele |
| LVKPSFYVYSRVKNL | E | FYVYSRVKN | YVYSRVKNL | Protective Allele |
| VYSRVKNLNSSRVPD | E | VKNLNSSRV | VKNLNSSRV | Protective Allele |
| SFYVYSRVKNLNSSR | E | YVYSRVKNL | YVYSRVKNL | Protective Allele |
| QFAYANRNRFLYIIK | M | YANRNRFLY | FAYANRNRF | Protective Allele |
| FIASFRLFARTRSMW | M | FRLFARTRS | RLFARTRSM | Protective Allele |
| TNILLNVPLHGTILT | M | ILLNVPLHG | ILLNVPLHG | Protective Allele |
| MADSNGTITVEELKK | M | ADSNGTITV | NGTITVEEL | Risk Allele |
| INWITGGIAIAMACL | M | ITGGIAIAM | ITGGIAIAM | Risk Allele |
| NWITGGIAIAMACLV | M | IAIAMACLV | ITGGIAIAM | Risk Allele |
| FKDQVILLNKHIDAY | N | ILLNKHIDA | VILLNKHID | Protective Allele |
| QVILLNKHIDAYKTF | N | ILLNKHIDA | NKHIDAYKT | Protective Allele |
| NFKDQVILLNKHIDA | N | VILLNKHID | FKDQVILLN | Protective Allele |
| LGTGPEAGLPYGANK | N | PEAGLPYGA | PEAGLPYGA | Risk Allele |
| GTGPEAGLPYGANKD | N | PEAGLPYGA | PEAGLPYGA | Risk Allele |
| YYLGTGPEAGLPYGA | N | TGPEAGLPY | PEAGLPYGA | Risk Allele |

immune response (LC6A20, LZTFL1, CCR9, FYCO1, CXCR6, and XCR1) and ABO blood group to severe COVID-19 disease, respectively [33]. These associations were replicated in subsequent studies [10, 34, 35]. Results of a genetic study of 2244 critically ill patients with COVID-19 in intensive care units across the UK identified associations on chromosomes 12q24.13, 19p13.2, 19p13.3, and 21q22.1, revealed the genes with innate immunity (IFNAR2 and OAS) and host-driven lung inflammatory injury (DPP9, TYK2, and CCR2) [8]. A large GWAS study from HGI, including three genome-wide association meta-analyses comprising 49,562 COVID-19 patients from 46 studies in 19 countries, identified 13 human genomic loci associated with infection or severe COVID-19, including TYK2, DPP9, and FOXP4 which corresponded to previously documented associations with lung or autoimmune diseases and inflammatory disorders [36]. Another GWAS study in Thai suggested a protective effect of IL17B on 5q32 against disease [37], and a recent release from IHG showed that MUC5B and ELF5 on chr11 might be associated with immune system regulation in the lungs in the development of COVID-19. Our study identified two loci on chromosomes 5q32, and 11p12 in European ancestry who reached the significance threshold of suggestive association with SARS-CoV-2 infection. We could not define a similar association in the AFR group, and this variation in gene-level may be responsible for the difference. Our findings, by sampling in an ethnically diverse region, as the first study to evaluate different populations in one GWAS study, confirm preliminary results on the genetic determinants of COVID-19 in a diverse population and further reflect the complexity of genetic factors involved in SARS-CoV-2 infection.

HLA molecules present antigens by binding to endogenous antigenic peptides (class I) or exogenous antigenic peptides (class II) and express them as peptide-MHC complexes on the surface of antigen-presenting cells. Previous studies from different countries have identified

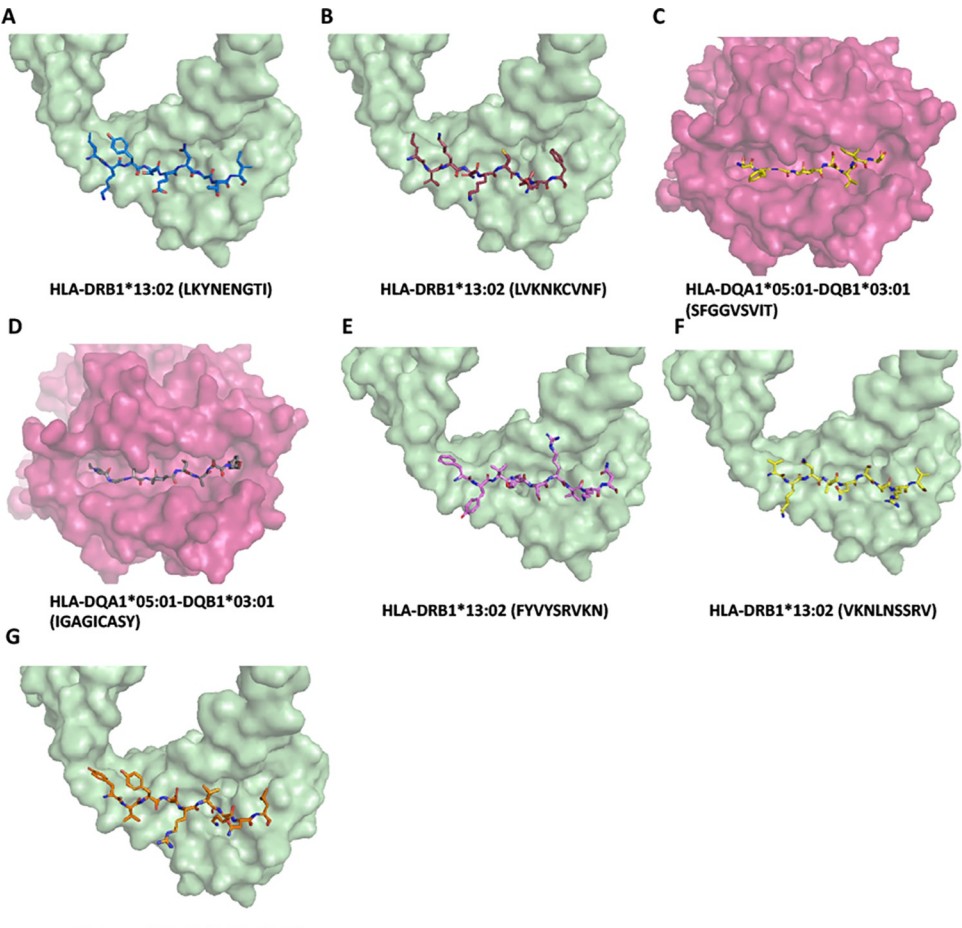

**Fig 6. Model of class II HLA molecule associated with protective (HLA-DRB1\*13:02) or risk (HLA-DQB1\*03:01) incidence of SARS-CoV-2 severity.** (A-B) HLA-DRB1\*13:02 binds to PRTFLLKYNENGTIT (LKYNENGTI), KKSTNLVKNKCVNFN (VKNKCVNF) and KSTNLVKNKCVNFNF (VKNKCVNF) derived from the S protein with affinity estimated Kd 14.09 nM, 63.11 nM, 41.78 nM respectively. (C-D) HLA-DQA1\*05:01-DQB1\*03:01 binds to ITPCSFGGVSVITPG (SFGGVSVIT), ECDIPIGAGICASYQ (IGAGICASY) and TPCSFGGVSVITPGT (SFGGVSVIT) derived from the S protein with affinity estimated Kd 51.75 nM, 28.32 nM, 52.35 nM respectively. (G-I) HLA-DRB1\*13:02 binds to LVKPSFYVYSRVKNL (FYVYSRVKN), VYSRVKNLNSSRVPD (VKNLNSSRV) and SFYVYSRVKNLNSSR (YVVYSRVKNL) derived from the E protein with affinity estimated Kd 223.87 nM, 135.65 nM, 162.90 nM respectively. Only the 9-mer core binding pepited were showed in the Figs. Peptides are shown as sticks modeled on the crystal structure of class II HLA molecues, different peptide backbones are represented by different colors.

multiple COVID-19 susceptibility-related alleles; for example, Wang et al. identified HLA-B\*15:27 alleles from a Chinese population [38], Yung et al. identified serotype B22 (HLA-B\*54:01, B\*56:01 and B\*56:04 alleles) from Hongkong Chinese population [39]. Our study identified that HLA-B\*27:05 in EUR ancestry and HLA-A\*02:01, -A\*33:01, -DPB1\*11:01 in AFR ancestry were associated with a decreased risk of SARS-CoV-2 positivity (OR <1), which provides additional clinical revealed alleles for a diverse population. During COVID-19, HLA appears to prevent or cause further disease progression through unknown mechanisms. One piece of evidence is that the low affinity of viral peptides to bind HLA can lead to severe disease and high-affinity binding, providing better protectivity. *In silico* prediction from Nguyen et al. identified HLA-A\*02:02, HLA-B\*15:03, and HLA-C\*12:03 have the strongest

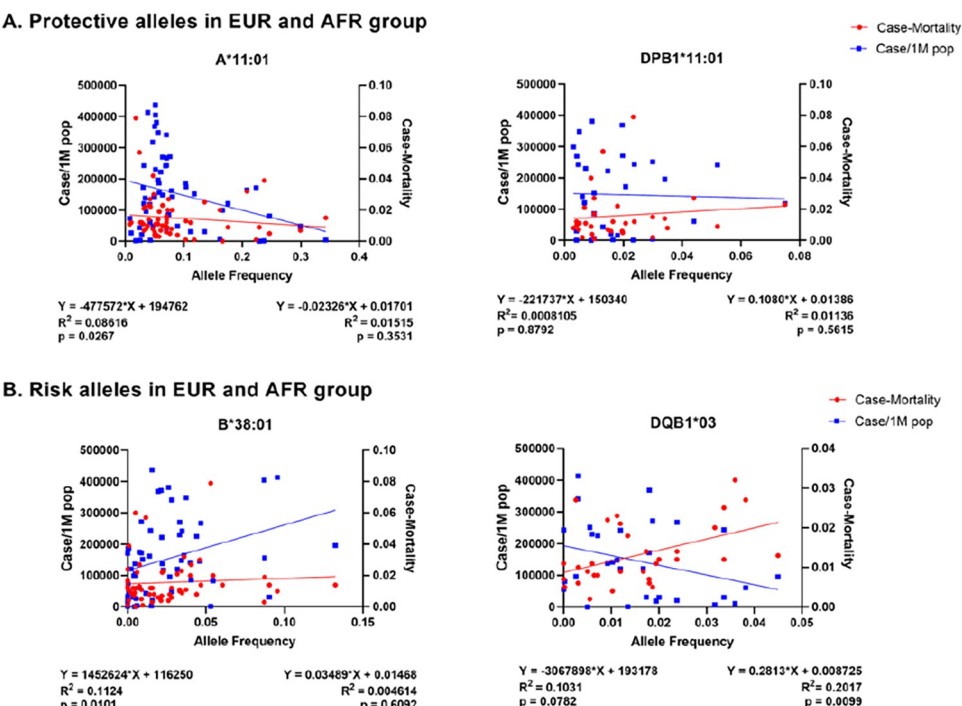

**Fig 7. Worldwide HLA allele frequency and COVID-19 case/1M population and case- mortaliy.** Association of protective and risk alleles in EUR and AFR cohort with worldwide case per one million population and case-mortality. Each dot represents a country plotted by average allele frequency (x-axis) with case-mortality (red, right y-axis) and case/1M population (blue, left y-axis). The equation and values below the image (left, case/1M population; right, case-mortality) show the quadratic equation for predicting trend of linear regression, the R-squared (coefficient of determination) and the two tailed p-value for the correlation analysis. (A) Association of proteictive alleles in EUR and AFR corhort. (B) Association of risk alleles in EUR and AFR corhort.

binding affinity of conserved peptides [11]. Amoroso et al. have shown that HLA-DRB1*08, which was predicted to bind SARS-CoV-2 peptides with low affinity, was correlated to mortality. To confirm this, a study with the Sardinian population found HLA-DRB1*08:01 allele only existed in hospitalized patients [16, 17]. A group from South Asia found HLA-B*35 was more among the mildly infected group than the fatal group and owned a high peptide loading capacity compared to other HLA-B proteins [14, 18]. Another evidence may come from cytokine storms, where an unbalanced immune response leads to higher morbidity, while a well-regulated immune response allows patients to recover more quickly. Although not studied in detail in COVID-19, it was shown in a previous study that class II MHC secretes different types of cytokines when binding to different peptides, thus triggering T cell differentiation in divergent ways (TH1/TH17 by DRB1*0401, risk; TH1/TH2 by DRB*0402, protective) in rheumatoid arthritis [40]. In our studies, we identified in EUR group HLA-C*12:03, -B*35, -B*38:01 were associated with an increased risk of SARS-CoV-2 severity (OR >1). And in the AFR group, HLA-DQB1*03 was associated with an increased risk of SARS-CoV-2 severity, while HLA-B*58, -DRB1*13:02, and -DQB1*06 were associated with a decreased risk of SARS-CoV-2 severity. HLA-C*12:03 and -B*35 identified in the severed EUR group were previously shown to be associated with less severity, and several class II MHC molecules were associated with increased or decreased severity in the AFR group, altogether suggesting a complicated disease progression mechanism that may involve in multiple alleles as well as host immune-regulated factors.

T cells have an important role in the outcome and maintenance of SARS-CoV-2 immunity normally during viral infection by recognizing viral antigens in short peptides present by HLA. A previous study identified epitope megapools (MPs) containing SARS-CoV-2 T cell epitopes derived from various viral proteins, which successfully induced viral-specific T cells responses in patients [41] and CoVac-1 vaccinated individuals [42]. CoVac-1 is a peptide-based vaccine candidate composed of predicted MPs (S, N, N, E, and ORF8); this prediction is based on the most prominent class I HLA-A and -B and class II HLA-DR alleles to protect a broad population. The success of phase 1 clinical trial of this vaccine demonstrates the potential and importance of epitope screening, as it can stimulate long-lasting T-cell immune responses in the fight against COVID-19. We performed *in silico* epitope mapping of the identified HLAs in different ancestry groups hoping that further experiments will lead to an affected-population-based understanding of immune defense mechanisms against SARS-CoV-2 and aid in the development of vaccines and immunotherapies. To understand the disease outcome in different ancestral origins, we applied a similar strategy; instead of utilizing common alleles, we used alleles associated with increased or decreased COVID-19 severity in both EUR and AFR groups (identified protective and risk alleles). We found the overall number of peptides that protective and risk alleles present differed, 46 vs. 151 in the EUR ancestry group and 248 vs. 206 in the AFR ancestry group. Due to the presentation difference in class I and class II MHC, the number could not simply explain the various disease outcome of different ancestry origins. In addition, the percentage of predicted stronger binders of protective and risk alleles is approximately 41% vs. 30% in the EUR ancestry group and 25% vs. 21% in the AFR ancestry group. The lesser portions to be presented in protective alleles in the AFR ancestry group might become an issue of ineffective immunity activation. Meanwhile, we observed some differences in the regions to be presented of protective and risk alleles, and we also observed that the E protein is only presented by class I MHC risk alleles in the EUR group, whereas it is presented by class II MHC protective alleles in the AFR group. This suggests that besides well-known RBD, certain regions of SARS-CoV-2 are also immunogenic, and peptides with insufficient immunogenicity should be considered for exclusion when developing vaccines or drugs for specific affected groups.

Age and pre-existing medical conditions such as hypertension, obesity, chronic lung disease, diabetes mellitus, and cardiovascular disease are associated with the severity and hospitalization rates of COVID-19 patients. In the COVID-NET catchment population which represents approximately 10% of the U.S. population with an equal ratio of male and female, 54% of COVID-19-associated hospitalizations occurred in males and 46% occurred in females [1]. These data suggest that males may be disproportionately affected by COVID-19 compared with females. Additionally, 59% of residents are white, 18% are black, and 14% are Hispanic; however, among 580 hospitalized COVID-19 patients with race/ethnicity data, approximately 45% were white, 33% were black, and 8% were Hispanic [1], suggesting that black populations might be unevenly affected by COVID-19 [4, 7]. Social-economics and access to quality healthcare are important factors in clinical outcomes. The epidemiology and the underlying reasons account for these differences need further investigation and beyond the scope of this study. Our data showed that the mean age and gender distribution of the EUR and AFR ancestry did not significantly differ in the case or control group. In conclusion, this study provides the first insight of group analysis regarding the effects of SARS-CoV-2 infection in different ancestry origins from the same region, including the susceptibility and disease severity. Due to the limited sample size, further validation is needed by *in-vitro* experiments. This study demonstrated that specific protective or less protective HLA alleles are predicted to present viral antigens differently, which could impact and contribute to the susceptibility and clinical outcome among populations.

## Supporting information

**S1 Fig. Quality control and population stratification in EUR ancestry.** (A) Differential principal components analysis based on variance; (B) Case (red) and control (blue) samples were plotted by PC1 and PC2; (C) 1000 genome reference population was used and plotted by PC1 and PC2, case and controls samples were overlaid with the reference population to show ethnic distribution.
(TIF)

**S2 Fig. Quality control and population stratification in AFR ancestry.** (A) Differential principal components analysis based on variance; (B) Case (red) and control (blue) samples were plotted by PC1 and PC2; (C) 1000 genome reference population was used and plotted by PC1 and PC2, case and controls samples were overlaid with the reference population to show ethnic distribution.
(TIF)

**S3 Fig. Distribution of allelic presentation of peptide across the SARS-CoV-2 structural for protective (HLA-B∗27:05) and risk (HLA-B∗38:01) alleles in EUR ancestry group.** Dark and light bars indicating the identified stronger (< 0.5%Rank) and weaker (< 2%Rank) binding 9 mer peptides, respectively. With green and red indicating protective and risk alleles association, respectively. Dashed lines are the selected top three peptides in each structural protein that were presented with the greatest variation in binding affinity, marked by the final tendency of alleles (green: protective allele; red: risk allele). The relative positions are arranged by successive structural proteins in the order of S, M, N and relative lengths as indicated in the bottom.
(TIF)

**S4 Fig. Model of class I HLA molecule associated with protective (HLA-B∗27:05) or risk (HLA-B∗38:01) incidence of SARS-CoV-2 severity.** (A-C) HLA-B∗27:05 binds to ARDLI-CAQK, RRARSVASQ and YRFNGIGVT derived from the S protein with affinity estimated Kd 1534 nM, 438.61 nM, 576.03 nM respectively. (D-F) HLA-B∗38:01 binds to DEDDSEPVL, EHVNNSYEC and QNAQALNTL derived from the S protein with affinity estimated Kd 12729.29 nM, 8508.57 nM, 12297.96 nM respectively.
(TIF)

**S1 Table. Samples information with age, gender and genetic associated ancestry.**
(XLSX)

**S2 Table. In silico binding prediction for protective (HLA-B∗27:05) or risk (HLA-C∗12:03) alleles in EUR group present structural protein of SARS-CoV-2 delta and omicron strain.**
(XLSX)

**S3 Table. In silico binding prediction for protective (HLA-B∗27:05) or risk (HLA-B∗38:01) alleles in EUR group present structural protein of SARS-CoV-2 original, delta and omicron strain.**
(XLSX)

**S4 Table. In silico binding prediction for protective (HLA-DRB1∗13:02) or risk (HLA-DQA1∗05:01-DQB1∗03:01) alleles in AFR group present structural protein of SARS-CoV-2 delta and omicron strain.**
(XLSX)

## Acknowledgments

We thanked Ms. Maria Cecilia Lopez from the UF Genetics Institute for performing the GWAS assays. We appreciated Dr. Patrick Concannon at the UF Genetics Institute for the insightful discussion.

## Author Contributions

**Conceptualization:** Yiran Shen, David A. Ostrov, Cuong Q. Nguyen.

**Data curation:** Bhuwan Khatri.

**Formal analysis:** Bhuwan Khatri, Danmeng Li, Christopher J. Lessard, Cuong Q. Nguyen.

**Funding acquisition:** Cuong Q. Nguyen.

**Investigation:** Yiran Shen, Cuong Q. Nguyen.

**Methodology:** Yiran Shen, Cuong Q. Nguyen.

**Resources:** Santosh Rananaware, Piyush K. Jain.

**Writing – original draft:** Yiran Shen, David A. Ostrov, Cuong Q. Nguyen.

**Writing – review & editing:** Yiran Shen, David A. Ostrov, Cuong Q. Nguyen.

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
