## [Decision Letter · Decision Letter 0]

19 Jul 2022

PONE-D-22-10827Ancestral origins are associated with SARS-CoV-2 susceptibility and protection in a Florida patient populationPLOS ONE

Dear Dr. nguyen,

Thank you for submitting your manuscript to PLOS ONE. After careful consideration, we feel that it has merit but does not fully meet PLOS ONE’s publication criteria as it currently stands. Therefore, we invite you to submit a revised version of the manuscript that addresses the points raised during the review process.

Both reviewers found the manuscript to be meritorious and well written, but raised couple questions regarding data presentation and interpretation. We look forward to receiving the revised version of your manuscript. Please ensure that your decision is justified on PLOS ONE’s publication criteria and not, for example, on novelty or perceived impact.

We look forward to receiving your revised manuscript.

Kind regards,

Agnieszka Rynda-Apple, Ph.D.

Academic Editor

PLOS ONE

Journal Requirements:

“CQN is supported financially in part by PHS grants DE028544 and DE028544-02S1 from the National Institute of Dental and Craniofacial Research. We thanked Ms. Maria Cecilia Lopez from the UF Genetics Institute for performing the GWAS assays. We appreciated Dr. Patrick Concannon at the UF Genetics Institute for the insightful discussion.”

“The funders had no role in study design, data collection and analysis, decision to publish, or preparation of the manuscript”

Reviewers' comments:

Reviewer's Responses to Questions

**Comments to the Author**

1. Is the manuscript technically sound, and do the data support the conclusions?

Reviewer #1: Yes

Reviewer #2: Partly

2. Has the statistical analysis been performed appropriately and rigorously? 

Reviewer #1: Yes

Reviewer #2: I Don't Know

3. Have the authors made all data underlying the findings in their manuscript fully available?

Reviewer #1: Yes

Reviewer #2: Yes

4. Is the manuscript presented in an intelligible fashion and written in standard English?

Reviewer #1: Yes

Reviewer #2: Yes

5. Review Comments to the Author

Reviewer #1: To reveal genetic determinants associated with COVID-19 susceptibility or severity in the EUR and AFR ancestral populations in Florida, the authors performed GWAS analysis on the two populations separately. They identified two loci on chromosomes 5q32 and 11p12 in the EUR ancestry which were associated with the COVID-19 susceptibility, and showed suggestive association with two genes, PPP2R2B at 5q32, and LRRC4C at 11p12. Using HLA imputation, they found that several class I MHC alleles in the EUR ancestry and several class I and II MHC alleles in the AFR ancestry were likely to be associated with disease susceptibility or severity. They also predicted SARS-CoV-2 peptides presented by protective and risk HLA alleles in EUR and AFR ancestries, suggesting the differential presentation of viral peptides in both ancestries. Lastly, it was suggested that extrapolation of protective and risk HLA alleles in the current study can be applicable in the worldwide population. This is a carefully done study and the findings are of considerable interest. However, I have raised several points which need to be clarified. These are given below.

Specific comments:

1. Fig. 3: To map which viral epitopes are presented by the protective and risk HLAs, the authors chose HLA-C*12:03 as a risk allele in the EUR ancestry group. However, I am very wondering how important HLA-C is in the HLA class I-related antigen presentation in comparison with HLA-A and HLA-B. I think the authors should have chosen HLA-B*38:01 rather than HLA-C*12:03 because HLA-B*38:01 showed a very big odds ratio (OR = 1639.00) (Table 2).

2. Page 19, lines 1-3 & Fig. 7B: The HLA-B*38:1 is not likely to show a positive or upward trend between the increase in the allele frequencies and rise in mortality (p=0.6092) (Figure 7B).　

3. Page 22, line 2 & Page 23, line 10: “the top three alleles” should be “the top three peptides”?

4. Page 23, line 5 & page 42, line 8, & Fig. 5: HLA- DQA1*05:01 is unlikely to be a risk allele because this allele is not found in Table 3.

Reviewer #2: The authors have presented a GWAS of COVID-19 positive individuals from Florida, a highly diverse state. Their analysis focused on collected nasal swabs and tracheal aspirates and looked at differences between individuals with European of African ancestry. The paper overall was well written but the approach is somewhat concerning.

Comments for the authors:

1. The determination of ancestry is not well defined, and supporting information was not available for review. The QC of the dataset was used "as previously described" but no ref is given and the information within the available methods is not sufficient. This needs to be described in more detail.

2. With a small sample size of 284 subjects, and no significant alleles determined for the AFR ancestry, it may be better to analyze the cohort as a whole, or breakdown between hospitalized and non-hospitalized subjects

3. The breakdown of ancestry to look at susceptibility to COVID-19 within this paper only considers the genetic component as analyzed and does not fully consider or discuss the social impacts that can impact disease exposure or severity, etc. These include, socio-economic status, health care access, education level etc. To make inferences based on ethnicity alone is questionable without fully addressing other contributing factors. This information may not be available for the studied cohort but should be considered when making conclusions about COVID-19 susceptibility and discussed further in the manuscript. The overall language used to compare different ancestries for COVID-19 susceptibility or severity needs to be carefully chosen and justified.

4. The authors state that they determined mild vs severe cases based on whether a nasal swab or tracheal aspirates, however this information only captures disease at the collection time. Do the authors a way to determine if the subjects that provided nasal swabs remained "mild" (ie no hospitalization")?

5. In addition, a table with more information about each subject within the different ancestries would also be helpful. Are those in one cohort older or younger than the other? Do they have any co-morbidities, especially subjects within the hospitalization group? Were any of the subjects vaccinated against COVID-19?

6. PLOS authors have the option to publish the peer review history of their article (what does this mean?). If published, this will include your full peer review and any attached files.

Reviewer #1: No

Reviewer #2: No

---

## [Author Response · Author response to Decision Letter 0]

7 Sep 2022

We would like to thank the reviewers and editorial team for your thoughtful and thorough comments on our manuscript. We have carefully considered each comment and have made changes and updates accordingly. Below is a detailed explanation of whether each change was incorporated. These changes are tracked in the revised manuscript; the number of lines marked up is based on the revised manuscript with track change.

Journal Requirements:

We have checked our manuscript and made changes to meet PLOS ONE's style requirements.

We have added "The written informed consent was collected from each participant prior to the start of the sampling" in the Method section (line 78). 

3. We note that the grant information you provided in the 'Funding Information' and 'Financial Disclosure' sections do not match.

We have corrected this issue in the resubmission. 

"CQN is supported financially in part by PHS grants DE028544 and DE028544-02S1 from the National Institute of Dental and Craniofacial Research. We thanked Ms. Maria Cecilia Lopez from the UF Genetics Institute for performing the GWAS assays. We appreciated Dr. Patrick Concannon at the UF Genetics Institute for the insightful discussion."

"The funders had no role in study design, data collection and analysis, decision to publish, or preparation of the manuscript"

We have deleted "CQN is supported financially in part by PHS grants DE028544 and DE028544-02S1 from the National Institute of Dental and Craniofacial Research" from the acknowledgment section. We have added the amended statement in the cover letter. 

Upon re-submitting your revised manuscript, please upload your study's minimal underlying data set as either Supporting Information files or to a stable, public repository and include the relevant URLs, DOIs, or accession numbers within your revised cover letter. For a list of acceptable repositories, please see http://journals.plos.org/plosone/s/data-availability#loc-recommended-repositories. Any potentially identifying patient information must be fully anonymized.

We have revised the "Data availability" with additional text "The data discussed in the study have been deposited in NCBI's Gene Expression Omnibus (GEO) and are accessible through GEO Series accession number GSE211972 (https://www.ncbi.nlm.nih.gov/geo/query/acc.cgi?acc=GSE211972)."

 

Reviewer #1

Specific comments:

1. Fig. 3: To map which viral epitopes are presented by the protective and risk HLAs, the authors chose HLA-C*12:03 as a risk allele in the EUR ancestry group. However, I am very wondering how important HLA-C is in the HLA class I-related antigen presentation in comparison with HLA-A and HLA-B. I think the authors should have chosen HLA-B*38:01 rather than HLA-C*12:03 because HLA-B*38:01 showed a very big odds ratio (OR = 1639.00) (Table 2).

We appreciate the reviewer's suggestion. The reviewer is correct that HLA-B*38:01 showed the highest odds ratio in the EUR ancestry. The main reason we chose HLA-C*12:03 to analyze was that HLA-C*12:03 has the lowest p-value, which we thought was more typical when considering potential molecular mechanisms in viral epitope mapping. We agree with the reviewer that HLA-B*38:01, which has a very big odds ratio, should be examined as well. As recommended, we performed the epitope mapping analysis for HLA-B*27:05 (protective) and HLA-B*38:01 (risk) (see S3 and S4 Figs and S3 table for details). With the new data, we revised the result section by adding the following sentences (lines 379-384). 

"Considering that HLA-B*38:01 has a large odds ratio (OR = 1639.00), we also did the same analysis (S3 Table) (S3 and S4 Figs) with HLA-B*38:01 as a risk allele compared to HLA-B*27:05 (protective). There was no obvious difference in the pattern and type of antigen presentation preference between the two alleles, as the presentation of S proteins was concentrated in the S2 region, and both alleles were unable to present E proteins."

2. Page 19, lines 1-3 & Fig. 7B: The HLA-B*38:1 is not likely to show a positive or upward trend between the increase in the allele frequencies and rise in mortality (p=0.6092) (Figure 7B).　

We thank the reviewer for pointing out the inaccuracy of this description. We have made the following changes in lines 510-511.

"Although the trends were similar, this was not significant in the mortality correlation (p=0.6092) (Fig 7B)". 

3. Page 22, line 2 & Page 23, line 10: "the top three alleles" should be "the top three peptides"?

We thank the reviewer for pointing out this wording error. We checked and switched "alleles" to "peptides" in lines 121, 126, and 360. 

4. Page 23, line 5 & page 42, line 8, & Fig. 5: HLA- DQA1*05:01 is unlikely to be a risk allele because this allele is not found in Table 3.

Thanks for pointing out our confusing description. The reviewer is correct that HLA- DQA1*05:01 is indeed not a significant risk allele in AFR group. 

According to the NetMHCIIpan 4.0 server (https://services.healthtech.dtu.dk/service.php?NetMHCIIpan-4.0), the prediction based on HLA-DQ molecules requires both α and β chains, and the only significant allele in the AFR group is HLA- DQB1*03 :01 (β chains). To prevent misleading random selection, we chose HLA- DQA1*05:01 as the α-strand for prediction. The detailed reasons are as follows.

1. According to the allele frequency network database (http://allelefrequencies.net/default.asp), the haplotype frequency of HLA-DQA1*05:01-DQB1*03:01 is relatively high in the population (up to 34%).

2. HLA-DQA1*05:01 does appeared in our HLA imputation with OR = 2.555, and p = 0.2915 (not significant)

3. HLA-DQA1*05:01-DQB1*03:01 belongs to the HighQ-DQ datasthe et in NetMHCIIpan 4.0 server, which possesses high accuracy prediction of binding patterns.

To make it clear, we revised the result (lines 389-391) by adding: "Prediction based on HLA-DQ molecules requires both α and β chains, we chose HLA- DQA1*05:01 (not significant in HLA estimation) as the α-chain based on the overall haplotype frequency and predictive accuracy."

Reviewer #2: 

1. The determination of ancestry is not well defined, and supporting information was not available for review. The QC of the dataset was used "as previously described" but no ref is given and the information within the available methods is not sufficient. This needs to be described in more detail.

We apologize for not making these details clear in the main text. The process of determining ancestry and the quality control steps were written in the methods section, namely "Assessment of population stratification" and "Quality control” sections. The determination of ancestry by using genome reference SNP is a relatively standardized process in GWAS. We did not stratify the patient populations based on the base on the information provided by participants. 

As recommended by the reviewer, references and new descriptions were added to the manuscript to clarify the methodology. It can be found in lines 158-159, "Due to the ethnically diverse nature of Florida residents, we performed quality control of the dataset as described previously in methods[21]."

2. With a small sample size of 284 subjects, and no significant alleles determined for the AFR ancestry, it may be better to analyze the cohort as a whole, or breakdown between hospitalized and non-hospitalized subjects.

We agree with the reviewer that it is better to analyze the cohort as a whole. In fact, that was what we did when we first conducted the study. However, when we performed the quality control step principal component analysis, it showed a significant difference in PC1. Furthermore, when we mapped our samples to the 1000 genomic reference population, we saw two independent clusters on PCA, and a larger number of sample scatters that could not be genetically annotated (please see the accompanying figure below). This means that our samples are fundamentally different in terms of genetic background. Some of the previous studies we cited, such as [33], used the same method as we did to exclude non-European ancestry and even subdivided the population to 8,036 Spanish and 4,273 Italian. Given the diversity of the Florida population, we decided to classify by genetic background for the accuracy of our study, even though our sample size was small.

3. The breakdown of ancestry to look at susceptibility to COVID-19 within this paper only considers the genetic component as analyzed and does not fully consider or discuss the social impacts that can impact disease exposure or severity, etc. These include, socio-economic status, health care access, education level etc. To make inferences based on ethnicity alone is questionable without fully addressing other contributing factors. This information may not be available for the studied cohort but should be considered when making conclusions about COVID-19 susceptibility and discussed further in the manuscript. The overall language used to compare different ancestries for COVID-19 susceptibility or severity needs to be carefully chosen and justified.

We appreciate the reviewer's thoughtful insight. As recommended, we have revised the discussion with new references to address this issue. 

Lines 634-681: "Age and pre-existing medical conditions such as hypertension, obesity, chronic lung disease, diabetes mellitus, and cardiovascular disease are associated with the severity and hospitalization rates of COVID-19 patients. In the COVID-NET catchment population which represents approximately 10% of the U.S. population with an equal ratio of male and female, 54% of COVID-19-associated hospitalizations occurred in males and 46% occurred in females[1]. These data suggest that males may be disproportionately affected by COVID-19 compared with females. Additionally, 59% of residents are white, 18% are black, and 14% are Hispanic; however, among 580 hospitalized COVID-19 patients with race/ethnicity data, approximately 45% were white, 33% were black, and 8% were Hispanic[1], suggesting that black populations might be unevenly affected by COVID-19[4,7]. Social economics and access to quality healthcare are important factors in clinical outcomes. The epidemiology and the underlying reasons account for these differences need further investigation and beyond the scope of this study. Our data showed that the mean age and gender distribution of the EUR and AFR ancestry did not significantly differ in the case or control group. In conclusion, this study provides the first insight of group analysis regarding the effects of SARS-CoV-2 infection in different ancestry origins from the same region, including the susceptibility and disease severity. Due to the limited sample size, further validation is needed by in-vitro experiments. This study demonstrated that specific protective or less protective HLA alleles are predicted to present viral antigens differently, which could impact and contribute to the susceptibility and clinical outcome among populations."

4. The authors state that they determined mild vs severe cases based on whether a nasal swab or tracheal aspirates, however this information only captures disease at the collection time. Do the authors a way to determine if the subjects that provided nasal swabs remained "mild" (ie no hospitalization")?

This is an important point that the reviewer raised. In response to this comment, during the past two months, we have revised our IRB to determine the prospective clinical status of these donors. Furthermore, we needed to identify the comorbidities (in response to comment #5). This information would provide additional important insights into the study. Unfortunately, as quoted by the Biorepository, "No information on the setting where samples were collected was available. As a consequence, further analyses on patient hospitalization status, length of stay, mortality, and comorbidities were not possible". 

5. In addition, a table with more information about each subject within the different ancestries would also be helpful. Are those in one cohort older or younger than the other? Do they have any comorbidities, especially subjects within the hospitalization group? Were any of the subjects vaccinated against COVID-19?

We appreciate the reviewer's feedback. Per suggestion, Table S1 was created with all the available information provided to us and added to the manuscript as supporting information. We have revised the discussion with this information (lines 641-642). Regarding vaccination status, all control and positive samples were collected before the COVID-19 vaccine was available, and theoretically, they were not vaccinated. Regarding the comorbidities and hospitalization status, please see our previous response (comment #4).

---

## [Decision Letter · Decision Letter 1]

20 Sep 2022

PONE-D-22-10827R1Ancestral origins are associated with SARS-CoV-2 susceptibility and protection in a Florida patient populationPLOS ONE

Dear Dr. nguyen,

Thank you for submitting your manuscript to PLOS ONE. After careful consideration, we feel that it has merit but does not fully meet PLOS ONE’s publication criteria as it currently stands. Therefore, we invite you to submit a revised version of the manuscript that addresses the points raised during the review process.

We look forward to receiving your revised manuscript.

Kind regards,

Agnieszka Rynda-Apple, Ph.D.

Academic Editor

PLOS ONE

Journal Requirements:

Additional Editor Comments:

Thank you for submitting revised manuscript. Please address the new comment by Reviewer 2. Please also ensure that all figures are appropriately labeled. We look forward to receiving your revised manuscript. 

Reviewers' comments:

Reviewer's Responses to Questions

**Comments to the Author**

1. If the authors have adequately addressed your comments raised in a previous round of review and you feel that this manuscript is now acceptable for publication, you may indicate that here to bypass the “Comments to the Author” section, enter your conflict of interest statement in the “Confidential to Editor” section, and submit your "Accept" recommendation.

Reviewer #1: All comments have been addressed

Reviewer #2: All comments have been addressed

2. Is the manuscript technically sound, and do the data support the conclusions?

Reviewer #1: Yes

Reviewer #2: Yes

3. Has the statistical analysis been performed appropriately and rigorously? 

Reviewer #1: Yes

Reviewer #2: I Don't Know

4. Have the authors made all data underlying the findings in their manuscript fully available?

Reviewer #1: Yes

Reviewer #2: Yes

5. Is the manuscript presented in an intelligible fashion and written in standard English?

Reviewer #1: Yes

Reviewer #2: Yes

6. Review Comments to the Author

Reviewer #1: (No Response)

Reviewer #2: The authors have submitted a well revised manuscript that more clearly communicates their data.

However upon revising the submitted material, it was noted that the AFR control group only consisted of 11 subjects, while the EUR control group comprised 57 (Supp Table 4). Can the authors comment on why there are so few individuals in the ARF control group compared to the EUR group and does this affect any of the conclusions made about the data?

Also to note, the Supplemental tables file names are mislabeled.

7. PLOS authors have the option to publish the peer review history of their article (what does this mean?). If published, this will include your full peer review and any attached files.

Reviewer #1: No

Reviewer #2: No

---

## [Author Response · Author response to Decision Letter 1]

27 Sep 2022

We would like to thank the editor and reviewer again for the feedback. We have carefully considered your comments. Below is a detailed explanation of whether each change was incorporated. Additionally, we have made one correction regarding the number of peptides we screened with HLA-C*12:03 (changed from 46 to 151 peptides, line 271). Lastly, we have corrected a few grammatical errors in the revised manuscript. 

Reviewer #2: The authors have submitted a well revised manuscript that more clearly communicates their data. However, upon revising the submitted material, it was noted that the AFR control group only consisted of 11 subjects, while the EUR control group comprised 57 (Supp Table 4). Can the authors comment on why there are so few individuals in the ARF control group compared to the EUR group and does this affect any of the conclusions made about the data?

We appreciate the reviewer's concern. We sampled based on whether or not being diagnosed with COVID-19 and disease severity. The distribution in the control group matched the population distribution in Florida, as we discussed in the result section with the heading "GWAS analysis of COVID-19 patients with European and African ancestries", in which 53% of Floridians are White (Non-Hispanic), with 21.6% being White (Hispanic), Black or African American (Non-Hispanic) (15.2%). As presented, the case group had a slightly higher proportion of AFRs, which is consistent with our conclusion of potential genetic susceptibility. Since we took EUR and AFR separately for comparison with their respective control group, the impact on the result should be minimal. Further in vitro experiments are needed to confirm these findings as proposed in the discussion. 

Also to note, the Supplemental tables file names are mislabeled.

We thank the reviewer for pointing out the mislabeled file names. We have corrected the file names for all three supplementary tables.

---

## [Editor Report · Decision Letter 2]

12 Oct 2022

Ancestral origins are associated with SARS-CoV-2 susceptibility and protection in a Florida patient population

PONE-D-22-10827R2

Dear Dr. nguyen,

We’re pleased to inform you that your manuscript has been judged scientifically suitable for publication and will be formally accepted for publication once it meets all outstanding technical requirements.

Kind regards,

Agnieszka Rynda-Apple, Ph.D.

Academic Editor

PLOS ONE
---

## [Editor Report · Acceptance letter]

6 Jan 2023

PONE-D-22-10827R2 

Ancestral origins are associated with SARS-CoV-2 susceptibility and protection in a Florida patient population 

Dear Dr. Nguyen:

I'm pleased to inform you that your manuscript has been deemed suitable for publication in PLOS ONE. Congratulations! Your manuscript is now with our production department. 

Kind regards, 

on behalf of

Dr. Agnieszka Rynda-Apple 

Academic Editor

PLOS ONE